# Ultrathin Single-Walled Carbon Nanotube Surface Wave Absorbers for Terahertz Dielectric Waveguides

Nikolaos Xenidis [1] ✉, Mehrdad Rezaei Golghand [1], Nikita I. Raginov[2], Joachim Oberhammer[1], Dmitry V. Krasnikov[2], Albert G. Nasibulin [2] ✉ & Dmitry V. Lioubtchenko [1] ✉

Dielectric waveguides are an emerging platform for terahertz (THz) integrated circuits, but a key challenge for dense integration is the realization of terminations that enable both multi-port device characterization and elimination of electromagnetic interference. Here, we demonstrate a compact, broadband termination by coating silicon waveguides with ultrathin single-walled carbon nanotube (SWCNT) films. Fabricated via a floating-catalyst (aerosol) chemical vapor deposition process, film thicknesses vary from 2 to 53 nm and are characterized in 140-220 GHz. A 53 nm thick film introduces up to 47 dB of attenuation while maintaining over 20 dB reflection loss, confirming nearly reflection-free absorption. Shielding analysis shows absorption dominates over reflection, and a record specific shielding efficiency of $5.5 \times 10^9$ dB cm$^2$ g$^{-1}$ is achieved. This approach offers a footprint-efficient solution for high-density THz circuits without bulky, radiative terminations.

Dielectric waveguides have emerged as a leading platform in terahertz (THz) and sub-THz bands, owing to their intrinsically low-loss propagation, strong mode confinement, and compatibility with CMOS fabrication processes[1–3]. In particular, high resistivity float-zone intrinsic silicon is virtually lossless in the THz range, offers a high refractive index contrast ($n_{Si}$ = 3.42) for tight modal confinement and sharp bending, and can be patterned with nanometer precision using mature, high volume CMOS foundry techniques, enabling monolithic integration[4]. Moreover, the surface wave modes supported by open dielectric waveguides have a significant fraction of their electromagnetic energy residing in the evanescent field extending to their cladding, making them inherently sensitive to changes in the surrounding environment[5–7]. This property is ideal for on-chip sensing[8] and evanescent coupling[9]. Various all-silicon platforms have been proposed, with recent advances including silicon photonic crystal waveguides that exploit engineered bandgaps[10], effective medium cladding waveguides that tailor dispersion via subwavelength patterning[11], unclad waveguides that are entirely cladded by the ambient air[12], topological waveguides[13] that offer extremely sharp bends, and others. Together, these platforms pave the way towards fully integrated THz circuits[14,15].

A key challenge in complex THz networks on these platforms is the realization of well-matched terminations. Without proper termination, reflections at device discontinuities can cascade, degrading performance and altering the intended operation profile. Additionally, terminations are necessary for characterization of multi-port devices, such as directional couplers, where the unused ports must be terminated with matched loads. The conventional solution employed is usually an adiabatic or impedance-matched tapering of the waveguide cross-section to free space, gradually expanding the guided mode to induce radiation losses, operating as a dielectric rod antenna[16]. However, the efficiency of these structures depends on the length of the tapering, therefore consuming valuable chip area, and can radiate power into undesirable directions, complicating packaging, limiting integration density and creating electromagnetic pollution. Moreover, such tapers introduce points of mechanical fragility: the thin tips are prone to chipping or fracture during handling and packaging, which can permanently degrade circuit performance[17].

[1]Division of Micro and Nanosystems, KTH Royal Institute of Technology, Stockholm, Sweden. [2]Laboratory of Nanomaterials, Skolkovo Institute of Science and Technology, Moscow, Russia. ✉e-mail: xenidis@kth.se; a.nasibulin@skol.tech; dml@kth.se

Since many of the conventional microwave approaches for terminations do not scale down well to THz systems for a variety of reasons (such as thermal incompatibility of materials with standard CMOS processes, or fabrication complexity for realizing tapered geometries in microscales), a number of novel nanocomposites have gained significant interest for their absorption properties in the THz regime, such as graphene-based composites[18,19], Mxene composites[20,21], and carbon nanotubes (CNTs)[22,23], among others.

In particular, single-walled carbon nanotube (SWCNT) films, with their exceptional electrical conductivity and tunable optoelectronic properties, have been used in a variety of THz devices, such as varactors[24], modulators[25], detectors[26], polarizers[27], arrays of spiral zone plates[28], and others. Additionally, CNTs have demonstrated excellent absorption capabilities, from the microwave up to the visible regime[29–31]. In most of these studies however, the CNT-based absorbers are used either inside a hollow waveguide or in a free-space spectroscopic configuration. Studies in the infrared region have demonstrated interaction of SWCNT with guided waves in SWCNT-coated dielectric waveguides for saturable absorbers[32,33]. However, it is unclear whether these materials are effective absorbers of surface-wave modes propagating in a THz dielectric waveguide, where only part of the guided field overlaps with the films. Our approach targets directly these dielectric waveguide modes in low THz frequencies.

In this work, we introduce an ultracompact termination strategy based on polymer-free SWCNT thin films, produced via aerosol chemical vapor deposition (CVD) and dry-transferred onto silicon dielectric rod waveguides (DRW). SWCNT networks exhibit broadband absorption well into the THz regime, with tunable surface conductivities via film thickness control. By coating the waveguide with SWCNT films, we achieve strong, localized attenuation of the surface-guided wave without significantly altering the waveguide geometry or footprint. This approach preserves the high-index contrast and modal confinement of the silicon platform, while providing a broadband, reflection-free termination element that is orders of magnitude more compact than conventional tapers.

## Results and discussion
### Material and Structural Characterization
SWCNT thin films with optical transmittance values (at 550 nm) of 98%, 96%, 94%, 87%, 79%, and 60% were prepared. Hereafter, each sample will be referred to by its measured optical transmittance at 550 nm. According to[34] the thickness of the corresponding SWCNT films can be estimated as approximately 2, 4, 6, 14, 24, and 53 nm, respectively. Figure 1 presents representative structural characterization data drawn from typical SWCNT films in this series. The thin films contain predominantly single-walled carbon nanotubes as directly observed with transmission electron microscopy (TEM; number of walls, Fig. 1c) as well as ultraviolet/visible/near-infrared spectroscopy (UV-Vis-NIR; appearance of the transitions between van Hove singularities, Fig. 1a) and Raman spectroscopy (the presence of radial breathing modes (RBM) characteristic to SWCNT; Fig. 1b). Graphitic $sp^2$-carbon structure manifests itself via the presence of a $\pi$-plasmon (Fig. 1a) and a characteristic G mode (Fig. 1b), while the low intensity of a D mode (Fig. 1b; $I_G/I_D$=41) verifies low concentration of defects and the high-quality of the produced samples. The films are formed by individual SWCNTs and their agglomerates (bundles; Fig. 1c), while the catalyst remains within the film without any effect on transparency in the visible light range or the conductivity[35]. The individual nanotubes and their bundles form a randomly-oriented highly porous network, as revealed by scanning electron microscopy, (SEM; Fig. 1d) with uniform spatial distribution (Supplementary Fig. 1b). A more detailed structural characterization of the thin SWCNT films employed here has been carried out in[36,37].

## Electromagnetic Characterization
The magnitude of the measured S-parameters of the SWCNT-loaded DRWs are given in Figs. 2, 3, along with an unloaded DRW for reference. The unloaded DRW has an average insertion loss of 0.7 dB across the band, indicating good coupling with the feeding hollow waveguides. Adding the SWCNT films yields extra losses that increase monotonically with thickness, spanning from an average of 3 dB across the band for the thinnest SWCNT-98% (2 nm) sample to an average of 47 dB for the thickest SWCNT-60% (53 nm) sample of 6 mm in length. As expected, the losses also increase with longer SWCNT films, with the corresponding SWCNT-60% sample approaching the noise floor towards the upper half of the band.

Reflection loss (Fig. 3) of the SWCNT-loaded DRWs is very close to that of the unloaded silicon DRW, indicating a good impedance matching and therefore pure attenuation with no extra reflection back to the feeding waveguides, which is particularly important for any absorbing material. Small peaks in $|S_{11}|$ (e.g. sample SWCNT-87%, 12 mm long) are due to sub-optimal manual alignment between the DRW and the hollow metallic waveguide. The qualified bandwidth, defined as the frequency range for which $|S_{11}| < -10$ dB, covers the entire 140-220 GHz band. Overall, the S-parameters indicate no particular selectivity in frequency for the targeted range, displaying a fractional bandwidth of 44.4%.

The shielding efficiencies of the SWCNT samples are calculated according to[38]:

$$A_{\text{eff}} = (1 - |S_{11}|^2 - |S_{21}|^2)/(1 - |S_{11}|^2), \tag{1}$$

$$SE_{\text{R}}(\text{dB}) = -10\log_{10}(1 - |S_{11}|^2), \tag{2}$$

$$SE_{\text{A}}(\text{dB}) = -10\log_{10}(1 - A_{\text{eff}}), \tag{3}$$

$$SE_{\text{T}} = SE_{\text{R}} + SE_{\text{A}}, \tag{4}$$

and are given in Fig. 4. We observe the primary shielding mechanism to be the absorption, since $SE_{\text{R}}$ remains sufficiently low across the whole frequency band. Table 1 gives the average reflection and total shielding efficiencies across the band, showing that the average reflection shielding efficiency is between 0.004 dB and 0.05 dB for all samples, while that of an unloaded DRW is 0.02 dB. Additionally, the average across the band total shielding efficiency is above 50 dB and 60 dB for the SWCNT-79% and SWCNT-60% (12 mm long) films, respectively, indicating excellent shielding through absorption.

Similar lossy SWCNT film-covered dielectric waveguides have been proposed before in[39–41]. However, the attenuation was significantly lower than the one presented here. This is because the SWCNT films are now placed parallel to the **E**-field. Since the $E_{11}^y$ mode is excited, the main components of the **E**-field are $E_y$ and $E_z$ (with reference to the axis of Supplementary Fig. 1a), with $E_x$ being significantly weaker and localized at the corners of the waveguide. Therefore, strong surface currents $\mathbf{J}_s = \sigma_s \mathbf{E}_t$ are induced (where $\sigma_s$ the surface SWCNT complex conductivity and $\mathbf{E}_t$ the tangential electric field at the wide wall of the DRW where the SWCNT films are deposited). In order to calculate the surface conductivity of each sample, a perturbation approach might be followed[42]. Assuming a single-mode operation and that the loss is primarily ohmic and attributed to the presence of the SWCNT films, the latter can be approximated as a small perturbation to the otherwise lossless unloaded waveguide. Additionally, we assume that the SWCNT film sheet conductivity is uniform (which

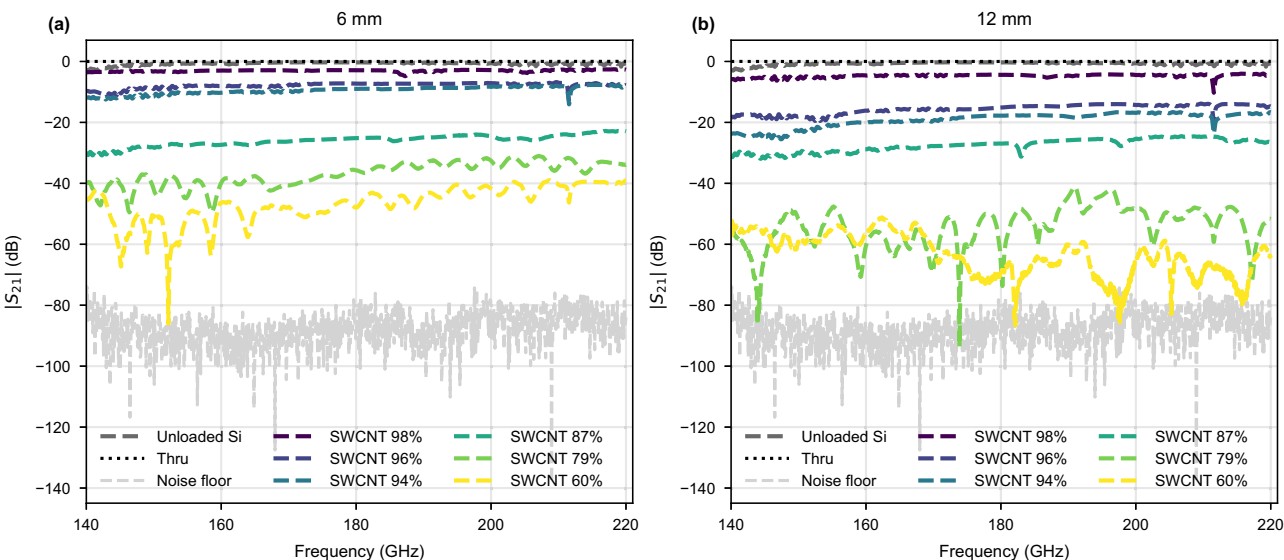

**Fig. 1 | Structural characteristics of the SWCNT films. a** UV-Vis-NIR spectrum (characteristic peaks are denoted). **b** Raman spectrum (characteristic modes are denoted). **c** Typical images of high-resolution TEM. **d** Typical SEM images.

**Fig. 2 | Transmission measurements of the SWCNT-loaded DRWs. a** $|S_{21}|$ for the 6 mm long samples. **b** $|S_{21}|$ for the 12 mm long samples. The black dotted line is thru after calibration (flanges of the frequency extenders connected together); dark grey is transmission of an unloaded DRW; light grey is the noise floor.

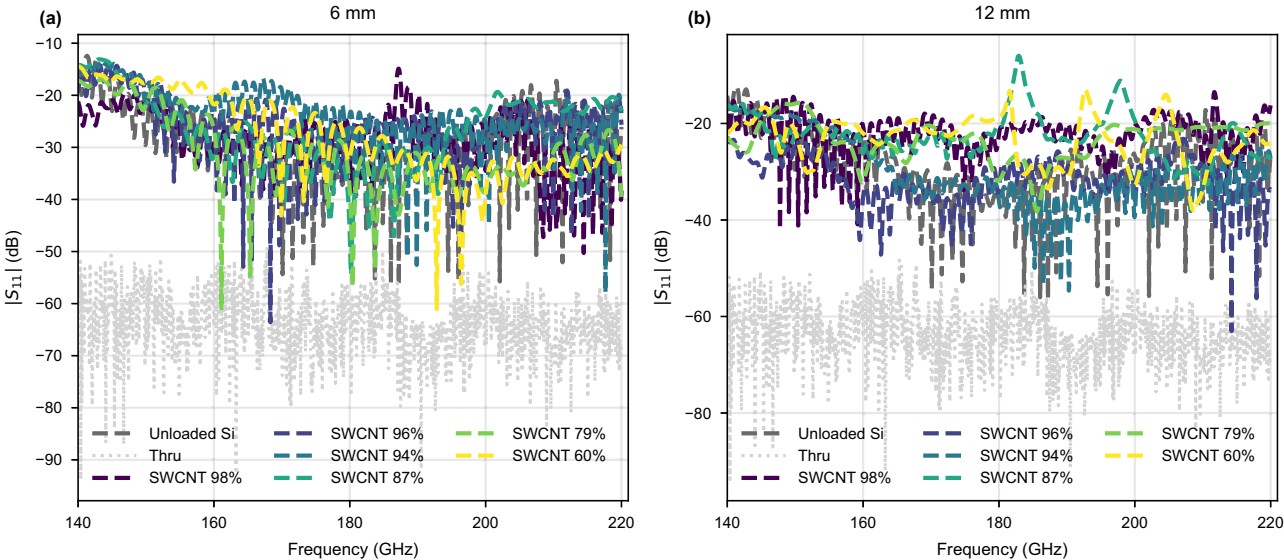

**Fig. 3 | Reflection measurements of the SWCNT-loaded DRWs. a** $|S_{11}|$ for the 6 mm long samples. **b** $|S_{11}|$ for the 12 mm long samples. The light grey line is baseline reflection after calibration by measuring a thru standard (flanges of the frequency extenders connected together); dark grey is the reflection coefficient of an unloaded DRW.

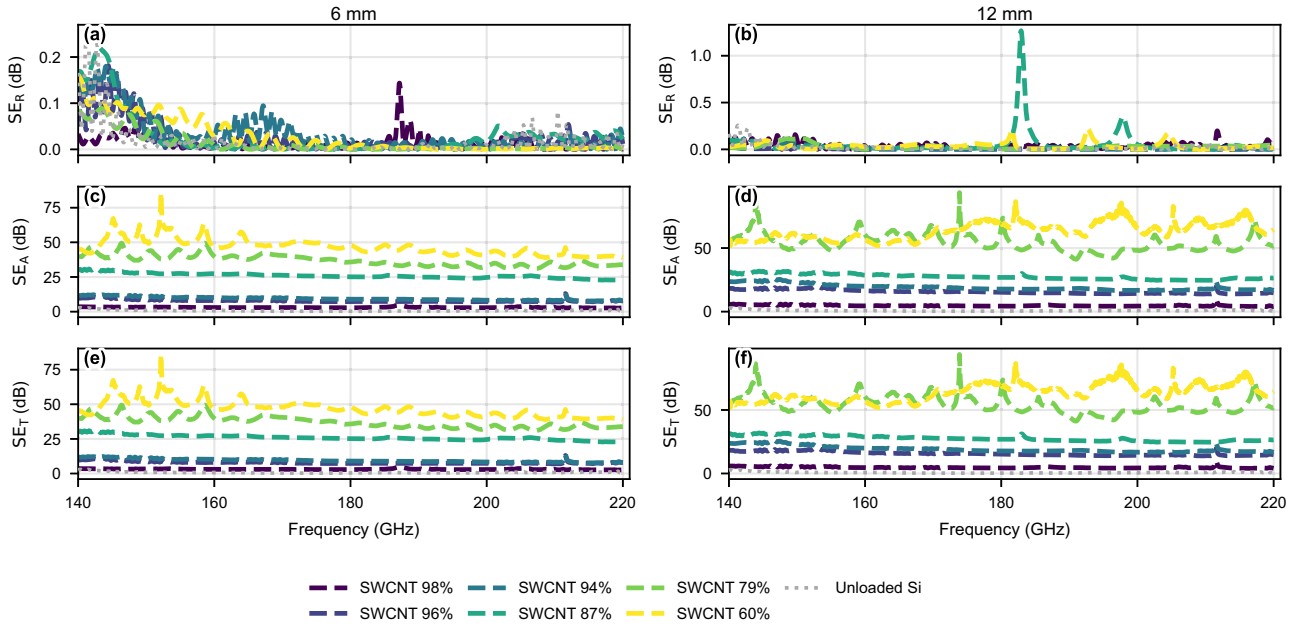

**Fig. 4 | Shielding efficiency components for the SWCNT-coated dielectric waveguides. a**, **b** Reflection component $SE_R$. **c,d** Absorption component $SE_A$. **e**, **f** Total shielding $SE_T$, for 6 mm (left column) and 12 mm (right column) samples over 140-220 GHz. With light grey the equivalent shielding efficiency of an unloaded silicon waveguide given for reference.

is confirmed by the uniformity of the samples, i.e. Supplementary Fig. 1b) and neglect its reactive part (assuming SWCNT films introduce only attenuation). The ohmic power loss per unit length due to the presence of the SWCNT films is given by[42]:

$$P_L = \frac{R_s}{2} \int_C |\mathbf{J}_s|^2 dl, \tag{5}$$

where $R_s$ is the sheet resistance of the SWCNT films and the contour integral runs across the SWCNT films in a transverse cross-section of the waveguide. Since power loss per unit length is $P_L = 2\alpha P_0$, with $P_0$ the excitation power and $\alpha$ the attenuation constant, then

$$R_s = \frac{\int_C |\mathbf{E}_t|^2 dl}{4\alpha P_0}, \tag{6}$$

where the tangential **E**-field component is approximated by that of the unloaded DRW. The attenuation constant can be calculated experimentally by $\alpha = \frac{1}{2L} \ln \frac{1 - |S_{11}|^2}{|S_{21}|^2}$ (Np/m), where $L$ the length of the SWCNT film along the propagation direction.

The tangential **E**-field component of the unloaded DRW was obtained through full-wave simulations in CST Microwave Studio, and then the extracted sheet resistance was used to simulate a surface impedance boundary on the top of the DRW. Polynomial fitting was used to fit the extracted sheet resistance to the boundary values in CST. In these simulations, the transitions from the hollow metallic waveguides of the frequency extenders to the DRW were included. Figure 5a, b shows the simulated *S*-parameters for 4 representative

### Table 1 | Summary of average reflection (SE_R) and total (SE_T) shielding efficiencies over 140–220 GHz for the 6 mm-long and 12 mm-long samples, together with the estimated thickness of each SWCNT film

| Optical transmittance (%) | Film thickness (nm) | SE_R (dB) | | SE_T (dB) | |
|---|---|---|---|---|---|
| | | 6 mm | 12 mm | 6 mm | 12 mm |
| 98 | 2 | 0.01 | 0.04 | 3.04 | 4.74 |
| 96 | 4 | 0.02 | 0.004 | 7.88 | 15.61 |
| 94 | 6 | 0.03 | 0.02 | 9.45 | 19.17 |
| 87 | 14 | 0.03 | 0.05 | 25.94 | 27.5 |
| 79 | 24 | 0.01 | 0.03 | 37.05 | 54.40 |
| 60 | 53 | 0.03 | 0.03 | 46.75 | 63.66 |
| Unloaded Si (reference) | | 0.02 | | 0.75 | |

For reference, the values of the unloaded silicon waveguide are also included.

samples (SWCNT-98%, SWCNT-96%, SWCNT-94% and SWCNT-79%, 6 mm long) next to the corresponding measured ones. The extracted THz sheet resistance used for the simulations is also given in Fig. 5c, while Fig. 5d shows the DC sheet resistance, measured using a 4-probe method. Although the simulated *S*-parameters follow the same frequency dependence and relative ordering of the samples, they overestimate the average (across the band) insertion loss by 0.8 dB, 1.8 dB, 2.2 dB and 6.1 dB for the samples SWCNT-98%, SWCNT-96%, SWCNT-94% and SWCNT-79%, respectively. Naturally, the perturbation approach is more accurate for the thinnest samples and less accurate for the thicker ones. Figure 6 shows the simulated electric field magnitude propagating across the loaded DRWs for the same samples, illustrating the attenuation in the SWCNT region.

The bulk density of the samples was evaluated before[43] using extremely precise balances of up to $1\,\mu$g precision and calculated as $1.6 \pm 0.5$ mg/cm³. The specific shielding efficiency (SSE), defined as SE/($\rho t$), with $t$ the thickness and $\rho$ the density of the material, is calculated as $5.5 \times 10^9$ dB cm² g⁻¹, an unprecedented level showcasing the excellent performance of the ultrathin SWCNT absorbers. Table 2 shows the comparison of key performance figures between the absorber of this work (with reference to the SWCNT-60%, 6 mm long sample) and other selected state of the art shielding materials reported in the literature.

## Methods
### Sample preparation
Thin films of single-walled carbon nanotubes were produced using a specific case of a floating catalyst CVD with an extreme dilution of a

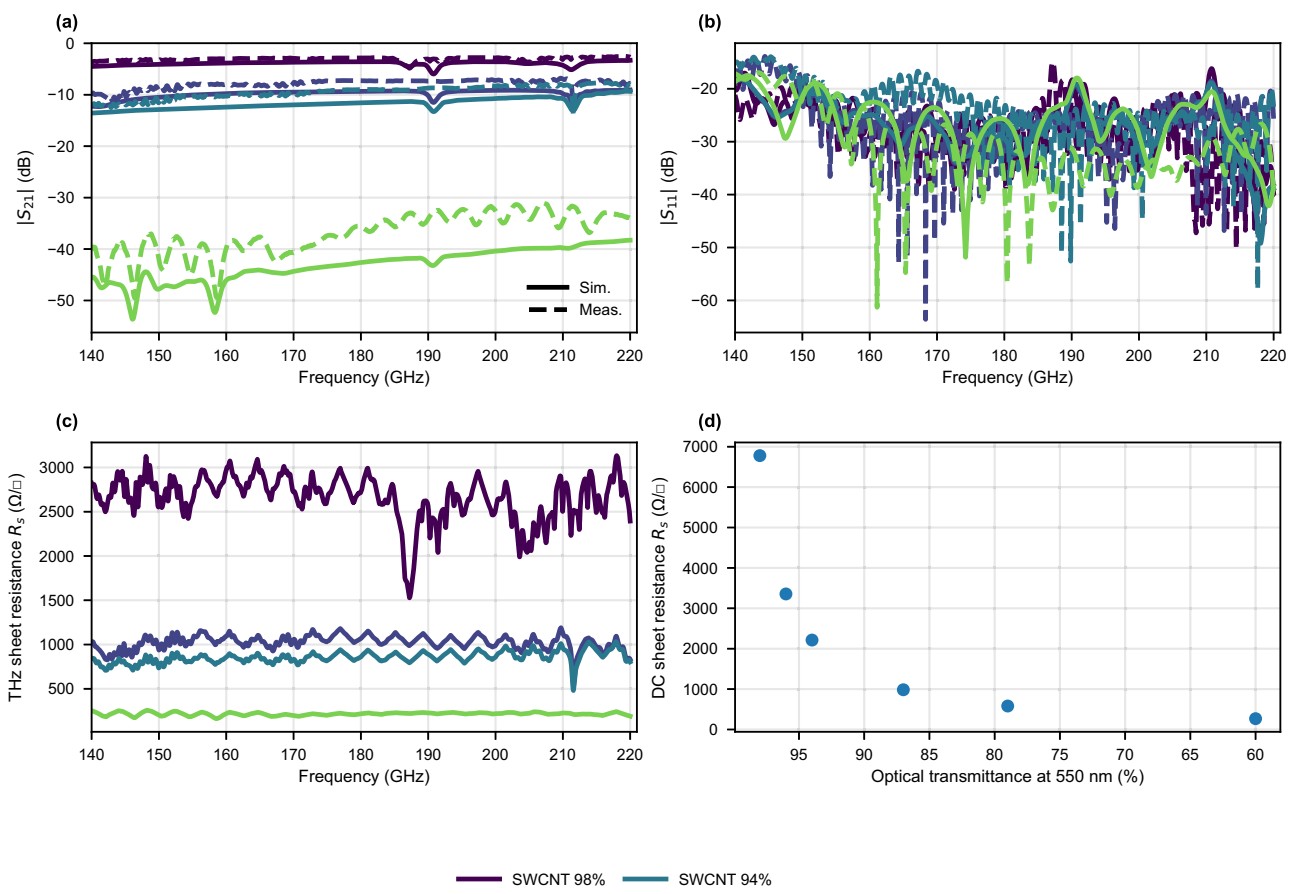

**Fig. 5 | Measured and simulated scattering parameters and sheet resistance of SWCNT-coated waveguides. a** Measured (dashed lines) and simulated (solid lines) $|S_{21}|$ for the samples SWCNT-98%, SWCNT-96%, SWCNT-94%, SWCNT-79% of 6 mm length. **b** Measured and simulated $|S_{11}|$ for the same samples. **c** The corresponding THz sheet resistance extracted using a perturbation approximation. **d** Measured DC sheet resistance.

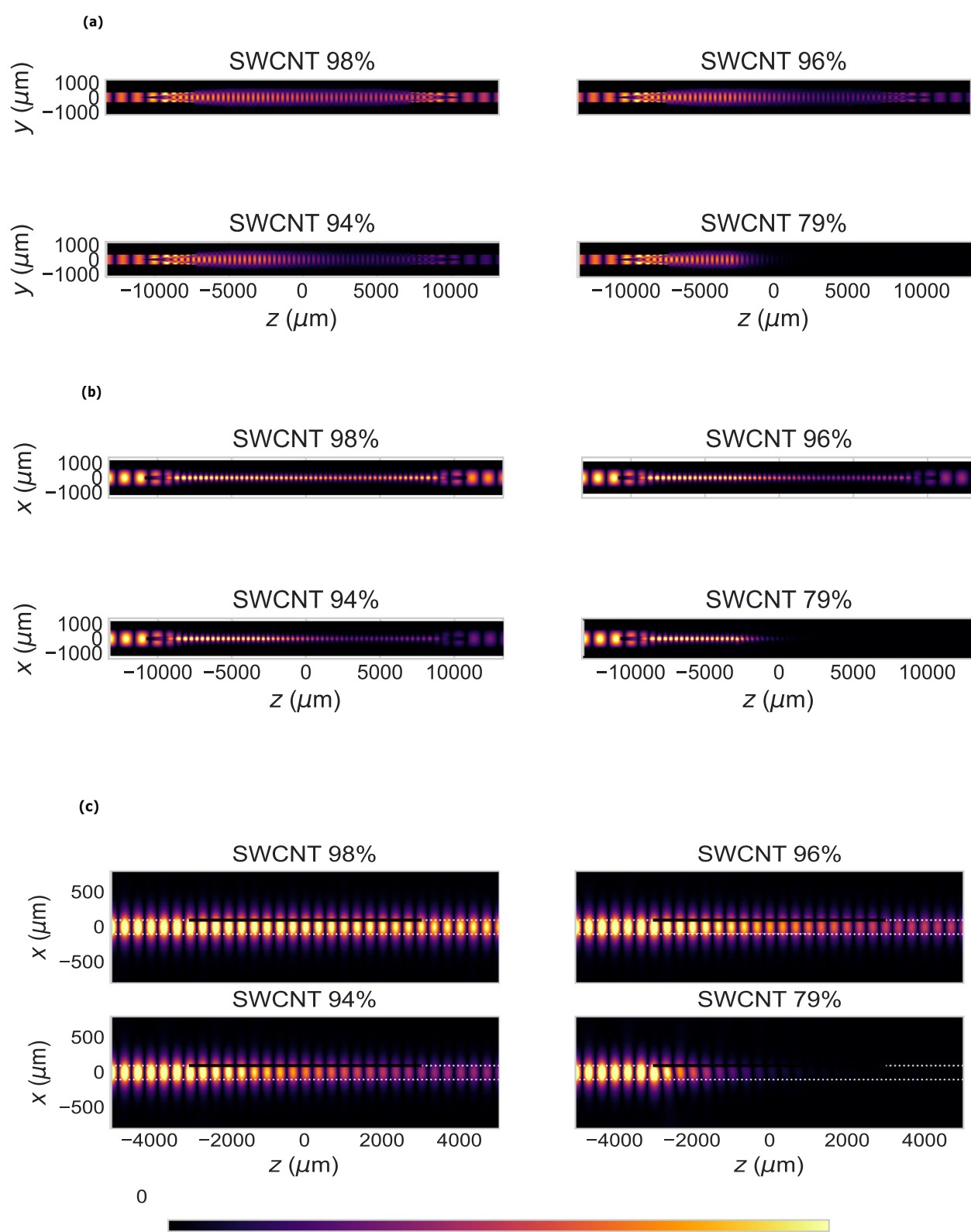

**Fig. 6 | Simulated electric field magnitude for wave propagation in the DRWs loaded with samples SWCNT-98%, SWCNT-96%, SWCNT-94%, SWCNT-79%.** **a** Top view (*yz*-plane). **b** Side view (*xz*-plane). **c** Close (*xz*-plane) view. White lines represent the boundaries of the DRWs and black lines the SWCNT film regions. The DRWs are excited through the hollow metallic waveguides. All field distributions correspond to 180 GHz.

**Table 2 | Comparison of shielding performance for various state-of-the-art absorbers**

| Reference | Freq. (GHz) | $SE_T$ (dB) | $SE_R$ (dB) | Length/Thickness (mm) | SSE (dB cm$^2$ g$^{-1}$) |
|---|---|---|---|---|---|
| 50 | 100–1600 | 40 | 0.078 | 2 | $1.8 \times 10^5$ |
| 51 | 200–1600 | 20 | 0.024 | 2 | $9.75 \times 10^4$ |
| 52 | 300–1600 | 60 | N/A | $2.13 \times 10^{-4}$ | $1.02 \times 10^5$ |
| 53 | 200–2000 | 60 | N/A | 0.033 | $3.0 \times 10^5$ |
| 54 | 200–2000 | 51 | 0.45 | 0.085 | $5.3 \times 10^4$ |
| This work | 140–220 | 46.75 | 0.03 | $6 / 53 \times 10^{-6}$ | $5.5 \times 10^9$ |

$SE_T$ and $SE_R$ are the average total and reflection efficiencies over the reported band.

catalyst–the aerosol CVD method[37]. The aerosol CVD reactor mainly comprised a quartz tube with a diameter of 72.5 mm and a length of 1,300 mm equipped with a three-zone furnace with a length of 1,000 mm to provide ca. 700 mm isothermal zone with a T of 854 °C. We employed the Boudouard reaction (CO disproportionation) on the surface of Fe-based nanoparticles to provide a catalytic growth of SWCNTs. The method provides only single-walled carbon nanotubes similar to the predominant growth of single-layer graphene[44] via the Boudouard reaction[45]. During the routine synthesis process, the flow of carbon monoxide (99.999% Linde gas; serves as a carbon source and a carrier gas) was fed to the reactor via three routes: a part of a flow went through a cartridge with a vapor of a volatile catalyst precursor (ferrocene, $C_{10}H_{10}Fe$, 99%, Sigma Aldrich), the second part mixed with the first one to be fed via an injector to facilitate the catalyst activation[46], and, lastly, the third and the most dominant fraction of CO flow was mixed with carbon dioxide ($CO_2$, 99.995%) as a mild etching agent to enhance growth[36] was fed into the main quartz tube. When the flow enters the hot zone of the reactor, ferrocene's transformation into an aerosol of Fe-based catalytic particles is followed by SWCNT growth. Thin films of SWCNT were collected with a simple filtration of the nanotube-containing aerosol with a nitrocellulose membrane (HAWP, Merck Millipore) to obtain a thin film of randomly oriented nanotubes for further dry transfer to any substrate[47]. Thickness was controlled by adjusting both the collection time and the aerosol flow through the filter.

## Scanning Electron Microscopy

To investigate the SWCNT film morphology (spatial distribution, alignment, impurities), a Quattro S scanning electron microscope in secondary electron (SE) mode at a 2 kV accelerating voltage was used.

## Transmission Electron Microscopy

The morphology of SWCNT networks (the number of walls, their structure, and bundling degree) was observed with the FEI Tecnai G2 F20 transmission electron microscope. For the sample collection, a TEM lacey Cu-300 grid was placed on a filter, and nanotubes with the aerosol flow coming from the reactor outlet were deposited directly on a grid.

## Raman Spectroscopy

Raman spectroscopy was employed to assess the quality of SWCNT films. Thermoscientific DXRxi Raman Imaging microscope operated at an excitation wavelength of 532 nm and a laser power of 0.1 mW.

## UV-Vis-NIR Spectroscopy

Ultraviolet-Visible-Near-infrared spectroscopy served to assess SWCNT structure ($S_{11}$, $S_{22}$, $M_{11}$, etc. transitions between the van Hove singularities directly relate to diameter distribution), possible doping (suppression of the transitions), and film thickness (based on the

absorbance at 550 nm). Perkin Elmer spectrophotometer Lambda 1050 was used to register spectra in the range of 200–1800 nm with a spectral resolution of 2 nm.

## Electromagnetic Characterization

For electromagnetic characterization, a vector network analyzer (VNA, Rohde & Schwarz ZVA-24) was connected to two frequency extender units (Rohde & Schwarz ZC220) to up-convert the frequency to 140-220 GHz. Thru-Offset-Short-Match (TOSM) calibration[48] was used at the flanges of the frequency extenders to correct for systematic errors. The DRWs are carefully inserted into the hollow metallic waveguides of the extenders using computer-controlled 3-axis positioner stages with micrometer accuracy. Care has been taken to align the DRW into the hollow waveguide opening and minimize generation of higher order modes[49]. By placing the DRW with their wide side parallel to the **E**-field of the hollow waveguide, the fundamental mode $E_{11}^y$ is excited[7]. A low-permittivity foam was used as a supporting structure to hold the DRW parallel to the feeding waveguides. Supplementary Fig. 1a shows a schematic of the DRW with the SWCNT layer on top, while Supplementary Fig. 1d shows the actual measurement setup in the lab.

The DRWs are 1 mm wide and were etched on a 200 $\mu$m high-resistivity float-zone silicon wafer using standard deep reactive ion-etching (DRIE) process. A 6 mm-long coupling spike is employed on both ends of the DRW in order to couple the THz waves from the hollow metallic waveguides of the frequency extenders. In total, the DRWs are 32 mm-long.

DC sheet resistance was measured with a Jandel RM3000 four-point probe station operated in current-source, voltage-sense mode. For each film, measurements were performed at least three times and the reported values are the average. The drive current was adjusted between 0.01 and 1 mA depending on the film conductivity.

## Data availability

The data generated in this study have been deposited in the figshare repository under accession code: https://doi.org/10.6084/m9.figshare.30391714. Simulation files can be obtained by contacting the corresponding authors.

## Code availability

Scripts used for processing and plotting the data make use of open-source Python libraries, and can be obtained from the corresponding authors upon request.

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

## Acknowledgements

We thank Cecilia Aronsson for the fabrication of the dielectric waveguides. N.X., M.R.G., D.L., and J.O. acknowledge the Swedish Foundation for Strategic Research (Project Number CHI19-0027). A.G.N. acknowledges Russian Science Foundation (Project Number 22-13-00436-П, synthesis and characterization of SWCNTs). This research made use of scikit-rf, an open-source Python package for RF and microwave applications.

## Author contributions

N.X. conceived the idea, curated the data and wrote the main manuscript. M.R.G. performed the THz measurements. D.K. and N.R. performed material synthesis and structural characterization. D.K. wrote the material and structural characterization part of the manuscript. A.N., J.O., D.L. supervised the project, reviewed the manuscript, acquired funding and provided resources.

## Funding

## Competing interests

The authors declare no competing interests.
