## [Transparent Peer Review file · Nature Communications]

Ultrathin Single-Walled Carbon Nanotube Surface Wave Absorbers for Terahertz Dielectric Waveguides

Corresponding Author: Mr Nikolaos Xenidis

Version 0:

Reviewer comments:

Reviewer #1

(Remarks to the Author)

The manuscript "Ultrathin Single-Walled Carbon Nanotube Surface Wave Absorbers for Terahertz Dielectric Waveguides" presents a novel method for terminating THz dielectric waveguides by coating their surface with ultrathin SWCNT films. The results demonstrate the high efficacy of the proposed approach and significantly extend the application domain of SWCNT films. I recommend this manuscript for publication after minor revisions, as outlined below.

General Comments

- 1) In the Introduction, the authors address the problem of THz waveguide terminations and discuss the limitations of geometrical tapering as the conventional solution. For broader contextualisation, it is recommended that the authors include a brief review of alternative absorbing materials that have been employed for THz terminations. This would help to better underscore the advantages of thin SWCNT films, particularly in terms of absorption efficiency, compactness, and integration compatibility.
- 2) Lines 180–181: "Interestingly, the 12 mm-long SWCNT60% and SWCNT79% samples have a similar attenuation profile across the band, indicating a saturation effect." Could this observation be a result of experimental uncertainty, given the very low signal amplitudes and the correspondingly high noise levels?
- 3) Fig. 3. Despite the anomaly of the 14 nm-thick film, the trend of increasing reflection with thicker SWCNT films can be observed. Could the authors comment on this? Might this suggest the existence of an optimum CNT film thickness for maximum shielding effectiveness (SE)?
- 4) For completeness, the authors are encouraged to include the data for the unloaded dielectric rectangular waveguide (DRW) as a reference in Fig. 4 and Table 1, where appropriate.
- 5) Fig. 5. It is notable how the ratios between the sheet resistance values derived from simulations compare with the ratios of DC sheet resistances. Could the authors provide the measured DC sheet resistance values for their films?
- 6) The study appears to employ only pristine (undoped) SWCNT films. Since the shielding mechanism relies on electrical conductivity, do the authors anticipate that doping could further enhance the SE of the SWCNT films?

Reviewer #2

(Remarks to the Author)

The paper describes carbon nanotube absorbers for dielectric (Si) waveguides in the 140-220 GHz band. The results appear to show efficient absorption with little reflection over the full measurement bandwidth.

The findings are reasonable and make sense, but I do not believe the work as presented is fitting for Nature Communications. Dielectric THz waveguides are a fairly niche area of technology, so something like IEEE Transactions on Terahertz Science and Technology may be a better fit.

Some issues to address in future iterations:

1. The authors claim, or at least suggest, ultra-broadband performance on line 210 based on the absence of frequency-selective absorption. That may be the case if the waveguides are scaled with frequency to maintain single-mode operation, but it strikes me as plausible that at higher frequencies where the given waveguides become multimodal that the absorption may degrade, since the modes become more confined to the Si core.
2. The reflection results from the 14nm sample are very odd, and the authors claim this could be due to suboptimal alignment or experimental error. I suggest discarding these results, since the errors are almost certainly due to errors rather than intriguing physics. As it is now, those results hinder the readability of the figures.
3. The paper can be hard to follow at times. The samples are labelled by their optical transmission, which are mapped to approximate thicknesses. They are then referred to in text by the approximate thicknesses, but labelled in figures by their optical transmissions. So on this line "The 14 nm thick film has higher reflections than the other samples for both 6 mm and 12 mm lengths" the reader needs to look at what optical transmission those correspond to first, then link that to the legend in the figures.
4. The Methods sections describes a range of metrology conducted (SEM, TEM, Raman spectroscopy), without any shown results. Taking for example Raman spectroscopy, how was this used to assess film quality? This would be useful in supplementary information.

Version 1:

Reviewer comments:

Reviewer #1

(Remarks to the Author)

The revised manuscript now appears to be ready for publication

(Remarks on code availability)

Reviewer #2

(Remarks to the Author)

I find the edits made by the authors to be satisfactory, and I have no further concerns on the manuscript. I appreciate them pointing out literature for this growing field, and would recommend this for publication in this journal.

(Remarks on code availability)

Responses to Reviewers' Comments for Manuscript NCOMMS-25-46006

**Ultrathin Single-Walled Carbon Nanotube
Surface Wave Absorbers for Terahertz
Dielectric Waveguides**

Addressed Comments for Publication to

by

Nikolaos Xenidis, Mehrdad Rezaei Golghand, Nikita I. Raginov, Joachim
Oberhammer, Dmitry V. Krasnikov, Albert G. Nasibulin and Dmitry V.

Lioubtchenko

Authors' Response to Reviewer 1

General Comments. The manuscript “Ultrathin Single-Walled Carbon Nanotube Surface Wave Absorbers for Terahertz Dielectric Waveguides” presents a novel method for terminating THz dielectric waveguides by coating their surface with ultrathin SWCNT films. The results demonstrate the high efficacy of the proposed approach and significantly extend the application domain of SWCNT films. I recommend this manuscript for publication after minor revisions, as outlined below.

Response: Thank you for your feedback and your valuable comments.

We have carefully addressed the rest of the issues according to your suggestions as follows.

Comment 1

In the Introduction, the authors address the problem of THz waveguide terminations and discuss the limitations of geometrical tapering as the conventional solution. For broader contextualisation, it is recommended that the authors include a brief review of alternative absorbing materials that have been employed for THz terminations. This would help to better underscore the advantages of thin SWCNT films, particularly in terms of absorption efficiency, compactness, and integration compatibility.

Response: Thank you for the comment.

While there is an ever-growing literature of THz absorbing materials for stealth and shielding applications, most of these studies are usually performed in a free-space spectroscopic configuration, with literature on integrated, device-level THz terminations being still sparse. For completeness, we have added the following paragraph to our introduction:

Since many of the conventional microwave approaches for terminations do not scale down well to THz systems for a variety of reasons (such as thermal incompatibility of materials with standard CMOS processes or fabrication complexity for realizing tapered geometries in microscales), a number of novel nanocomposites have gained significant interest for their absorption properties in the THz regime, such as graphene-based composites [1], [2], Mxene composites [3], [4], and carbon nanotubes (CNTs) [5], [6], among others.

- [1] H. Chen, W. Ma, Z. Huang, Y. Zhang, Y. Huang, and Y. Chen, “Graphene-based materials toward microwave and terahertz absorbing stealth technologies,” *Advanced Optical Materials*, vol. 7, no. 8, p. 1 801 318, 2019.
- [2] J. Campion et al., “Ultra-wideband integrated graphene-based absorbers for terahertz waveguide systems,” *Advanced Electronic Materials*, vol. 8, no. 9, p. 2 200 106, 2022.
- [3] W. Shui et al., “Ti₃C₂T_x mxene sponge composite as broadband terahertz absorber,” *Advanced optical materials*, vol. 8, no. 21, p. 2 001 120, 2020.
- [4] V. V. Starchenko et al., “Electrochemically and optically-switched terahertz electromagnetic interference shielding using mxenes,” *Physical Review Materials*, vol. 9, no. 7, p. 074 008, 2025.
- [5] D. Xiao et al., “Flexible ultra-wideband terahertz absorber based on vertically aligned carbon nanotubes,” *ACS applied materials & interfaces*, vol. 11, no. 46, pp. 43 671–43 680, 2019.
- [6] P. A. Drozd et al., “Highly efficient absorption of thz radiation using waveguide-integrated carbon nanotube/cellulose aerogels,” *Applied Materials Today*, vol. 29, 2022.

Comment 2

Lines 180–181: “Interestingly, the 12 mm-long SWCNT60% and SWCNT79% samples have a similar attenuation profile across the band, indicating a saturation effect.” Could this observation be a result of experimental uncertainty, given the very low signal amplitudes and the correspondingly high noise levels?

Response: Thank you for the comment.

Of course this is plausible and in fact, this was our first thought. However, our instruments provide a theoretical noise floor of -90 dB, which for the most part is below the measured transmission levels across the band. To disambiguate this, we have repeated the measurements and measured the actual noise floor of our system. In Fig. 2 of the main manuscript, the measured noise floor is given, which is indeed around -90 dB, and was measured by placing a short calibration standard between the hollow metallic waveguide flanges to isolate them. For the 12 mm long samples (Fig. 2b of the main manuscript), the new measurements show that the insertion loss of samples SWCNT79% and SWCNT60% is mostly overlapping up to around 170 GHz (with a value of around 60 dB), and above that frequency the insertion loss of the two samples is resolved, with $|S_{21}|$ of the thickest sample SWCNT60% almost touching the noise floor at certain points. For this new plot, a new sample was prepared by depositing a new SWCNT60% film on a new silicon waveguide. We have also removed the phrase “indicating a saturation effect” to avoid further speculations beyond what the data shows, and substituted it with the following:

As expected, the losses also increase with longer SWCNT films, with the corresponding SWCNT60% sample approaching the noise floor towards the upper half of the band.

Comment 3

Fig. 3. Despite the anomaly of the 14 nm-thick film, the trend of increasing reflection with thicker SWCNT films can be observed. Could the authors comment on this? Might this suggest the existence of an optimum CNT film thickness for maximum shielding effectiveness (SE)?

Response: Thank you for the comment.

We have repeated the measurements for the anomalous samples in the updated Figs. 2-3 of the original manuscript. In these updated figures, we cannot really see any trend of increasing reflection with thicker SWCNT films, beyond variances due to experimental uncertainty. For reference, we include here Table 1 with average return loss ($= -|S_{11}|$, dB) across the band, for all samples. As one can see, there is no clear trend for the available data points; for example, sample SWCNT98% (12 mm long), which is the thinnest one, has a higher reflection than all other 12 mm long samples, on average. Please keep in mind that the measured reflection in this experiment is mostly due to fixtures and the mode conversion between the hollow metallic waveguide mode TE_{10} and the DRW mode E_{11}^y . This can also be seen in Fig. 3 of the revised original manuscript, where for reference, we give the reflection of the thru standard (connecting the flanges of the hollow metallic waveguides together directly), which is the baseline minimum reflection captured with our instrumentation. Even an unloaded silicon waveguide has significantly higher reflection compared to this baseline, and this is well known phenomenon in the literature (for example, in [7] a similar reflection profile can be seen). The topic of launching THz waves in dielectric waveguides is in fact an active field of research [8]. Also, please note that the small peaks in $|S_{11}|$ even in these updated figures (for example, in sample SWCNT87% of Fig. 3b in the revised manuscript) are almost certainly due to misalignment effects [9]. As evidence for this, we provide two simulations in Fig. 1. The full chain of transitions between hollow metallic waveguide - silicon waveguide - hollow metallic waveguide is simulated, but with a translational misalignment Δz along the axis of propagation ($\Delta z = 0$: silicon waveguide touches the walls of the hollow metallic

waveguides, see inset). It can be seen that for sub-optimal alignment, $|S_{11}|$ rises; the frequency of the first peak also coincides with the peak of SWCNT87% in Fig. 3b of the main manuscript. For non-standardized waveguides without packaging/interfaces for precise alignment, these phenomena occur often.

Table 1: Average return loss (RL) across the 140-220 GHz band for all samples.

Length	Sample	RL (dB)
6 mm	SWCNT 60%	27.69
6 mm	SWCNT 79%	30.52
6 mm	SWCNT 87%	26.50
6 mm	SWCNT 94%	23.94
6 mm	SWCNT 96%	25.77
6 mm	SWCNT 98%	28.25
12 mm	SWCNT 60%	22.79
12 mm	SWCNT 79%	23.34
12 mm	SWCNT 87%	22.53
12 mm	SWCNT 94%	29.93
12 mm	SWCNT 96%	31.45
12 mm	SWCNT 98%	21.97
-	Unloaded Si DRW	28.55

Figure 1: Effect of translational misalignment Δz between the hollow metallic waveguide (feed) and the silicon waveguide on $|S_{11}|$.

That being said, it is still very plausible that optimum(s) for the thickness exists (see also our discussion in Comment 6), although for the available datapoints, this optimum manifests itself in terms of insertion loss rather than return loss.

- [7] H. Lees, W. Gao, and W. Withayachumnankul, “All-silicon, low-cross-talk terahertz waveguide crossing based on effective medium,” *Optics Letters*, vol. 46, no. 21, pp. 5469–5472, 2021.
- [8] D. Headland and G. Carpintero, “Robust unclad terahertz waveguides and integrated components enabled by multi-mode effects and matched slot couplers,” *IEEE Transactions on Terahertz Science and Technology*, pp. 1–9, 2025. DOI: 10.1109/TTHZ.2025.3583912.
- [9] S. Smirnov, N. Xenidis, J. Oberhammer, and D. V. Lioubtchenko, “Generation of high-order modes in sub-thz dielectric waveguides by misalignment of the transition structure,” in *2022 IEEE/MTT-S International Microwave Symposium - IMS 2022*, 2022, pp. 479–482. DOI: 10.1109/IMS37962.2022.9865497.

Comment 4

For completeness, the authors are encouraged to include the data for the unloaded dielectric rectangular waveguide (DRW) as a reference in Fig. 4 and Table 1, where appropriate.

Response: Thank you for the comment.

We have updated Fig. 4 and Table 1 of our revised manuscript to include also data for an unloaded DRW as a reference. However, it should be kept in mind that this acts only as a comparison reference, since an unloaded waveguide is simply a transmission line and its role is not to attenuate or shield between its two ends.

Comment 5

Fig. 5. It is notable how the ratios between the sheet resistance values derived from simulations compare with the ratios of DC sheet resistances. Could the authors provide the measured DC sheet resistance values for their films?

Response: Thank you for the comment.

We agree it is informative to relate THz extracted sheet resistances to DC values. The four point probe measured DC sheet resistance for each sample is given in Fig. 2; these values were mapped to R_{90} , defined as the equivalent sheet resistance for a film with 90% transmittance at 550 nm: $R_{90} = -R_s \log_{10}(T)/\log_{10}(10/9)$, with T the optical transmittance (%) at 550 nm [10]. All samples clustered at $R_{90} = 1300 \pm 100 \Omega/\text{sq}$. This figure has been included in Fig. 5d of the revised main manuscript. We note however that measuring DC sheet resistance of SWCNT films is highly sensitive to probe pressure, the ambient environment, adhesion to the substrate and time/aging.

Figure 2: Measured DC sheet resistance for all samples.

- [10] I. V. Novikov et al., “Aerosol cvd carbon nanotube thin films: From synthesis to advanced applications: A comprehensive review,” *Advanced Materials*, p. 2413777, 2025.

Comment 6

The study appears to employ only pristine (undoped) SWCNT films. Since the shielding mechanism relies on electrical conductivity, do the authors anticipate that doping could further enhance the SE of the SWCNT films?

Response: Thank you for the comment.

Yes, doping can indeed enhance the conductivity 3-5 times at least. We here refer to [11], where the doping procedure was thoroughly studied. Nevertheless, not all doping methods are stable while effective and robust methods require another reactor [12]. This is why it is out of the present paper’s scope.

Moreover, it should be noted that although the findings of this study follow a monotonic behavior between the increase in conductivity and increase in losses, this trend cannot be generalized arbitrarily. Between the two extremes of zero conductivity (perfect electric

insulator) and infinite conductivity (perfect electric conductor), optimum(s) for the losses exist. For both of the extreme cases, only the phase constant will change with no ohmic losses occurring, so between these extremes there is a regime where loss is maximized. To further illustrate this point, we simulate a single silicon waveguide of the same dimensions as the ones in the original manuscript ($1 \text{ mm} \times 200 \text{ }\mu\text{m}$ cross section), loaded with a thin, 6 mm long ohmic sheet. For simplicity, a single sheet resistance value is considered across frequency in each case (i.e. no dispersion is considered here). As can be seen from Fig. 3a, for very high conductivity (small R_s) the insertion loss is low. Increasing R_s (decreasing conductivity) yields higher insertion loss, up until a point ($\sim 220 \text{ }\Omega/\text{sq.}$). Further increasing R_s yields lower insertion loss again. On the other hand, $|S_{11}|$ seems to have an asymptotic behavior with increasing R_s (Fig. 3b). The point is further illustrated in Fig. 3c, where the average across the band insertion and return loss are calculated for varying R_s . At least two local maxima can be identified in the neighborhood of 10-400 Ω/sq for insertion loss. Interestingly, the highest local maximum seems to occur around the neighborhood of our thickest sample (which has an R_s of about 150-230 Ω/sq as obtained by perturbative analysis).

(a)

(b)

(c)

Figure 3: Simulated $|S_{21}|$ (a) and $|S_{11}|$ (b) for a silicon rod waveguide loaded with a thin resistive film (6 mm long) of varying sheet resistance. Average insertion and return loss across the band for varying sheet resistance is also given in (c).

- [11] D. A. Ilatovskii, E. P. Gilshtein, O. E. Glukhova, and A. G. Nasibulin, “Transparent conducting films based on carbon nanotubes: Rational design toward the theoretical limit,” *Advanced Science*, vol. 9, no. 24, p. 2201673, 2022.
- [12] E. M. Khabushev et al., “High-temperature adsorption of nitrogen dioxide for stable, efficient, and scalable doping of carbon nanotubes,” *Carbon*, vol. 224, p. 119082, 2024.

Concluding Response. Thank you for your valuable comments and for helping us strengthen our manuscript.

Authors' Response to Reviewer 2

General Comments. The paper describes carbon nanotube absorbers for dielectric (Si) waveguides in the 140-220 GHz band. The results appear to show efficient absorption with little reflection over the full measurement bandwidth.

The findings are reasonable and make sense, but I do not believe the work as presented is fitting for Nature Communications. Dielectric THz waveguides are a fairly niche area of technology, so something like IEEE Transactions on Terahertz Science and Technology may be a better fit.

Response: Thank you for your feedback.

We appreciate the perspective that THz dielectric waveguides may appear specialized. However, THz technology is expanding across sensing, imaging, spectroscopy, communications, and other areas, and as we note in the introduction, a number of different THz dielectric waveguide platforms has been proposed in the literature recently, gaining significant interest, which is primarily motivated by the high losses of metals in these frequencies [1]. A practical limitation for multi-port device characterization of these devices is the absence of compact terminations. The only solution in THz frequencies is the use of radiative coupling spikes, while the matter of electromagnetic pollution and shielding of these open waveguides has not been addressed at all, to the best of our knowledge. Our research proposes an alternative that addresses both matters. We therefore believe the results are of interest to a broad THz and integrated optics audience and appropriate for Nature Communications.

- [1] D. Headland, M. Fujita, G. Carpintero, T. Nagatsuma, and W. Withayachumnankul, "Terahertz integration platforms using substrateless all-silicon microstructures," *APL Photonics*, vol. 8, no. 9, 2023.

Comment 1

The authors claim, or at least suggest, ultra-broadband performance on line 210 based on the absence of frequency-selective absorption. That may be the case if the waveguides are scaled with frequency to maintain single-mode operation, but it strikes me as plausible that at higher frequencies where the given waveguides become multimodal that the absorption may degrade, since the modes become more confined to the Si core.

Response: Thank you for the comment.

This is a very valuable observation and we are very aware of this phenomenon. Indeed, operation in higher frequencies will translate to tighter mode confinement and a smaller part of the guided field will spread out to the cladding. If we assume truly multimodal propagation, then the total guided power is distributed among the number of propagating modes. While the fundamental mode is confined tighter inside the Si core with increasing frequency, the higher order modes are always less tightly confined. It is therefore plausible that the net attenuated power (for the sum of all modes) remains roughly the same.

On the other hand, one might assume that the waveguide operates in the overmoded regime (meaning that the frequency is sufficiently high) but still only the fundamental mode is of concern. In this case, since the absorber is of surface type and not volumetric, the most important factor is the tangential field at the discontinuity between core and cladding. Fig. 4a shows the (normalized to the global maximum) electric field distribution across the narrow wall of the silicon rod waveguide for frequencies spanning 140-500 GHz, where indeed it can be seen that the evanescent field reduces, and so does its value right at the boundary where the SWCNT films sit. Fig. 4b shows a time-domain simulation of a silicon rod waveguide with the same dimensions as the ones in our manuscript, loaded with a thin resistive film of $R_s = 200 \Omega/\text{sq}$. (constant across frequency). Here it can be seen clearly that insertion loss indeed reduces with increasing frequency, as was correctly pointed out. However, even in this simplified case, insertion loss remains significant for a frequency range spanning at least 3 bands (WR-5.1, WR-3.4, WR-2.2). Of course, this

should be considered as supportive evidence only, and additional experiments (which are the scope of a future work) will be the ultimate source of truth. Nevertheless, we have replaced the following statement:

~~and therefore the absorption bandwidth of the material can be extended to other frequency bands, making it ultra-broadband.~~

with a more conservative statement:

Overall, the S -parameters indicate no particular selectivity in frequency for the targeted range, displaying a fractional bandwidth of 44.4%.

(a)

(b)

Figure 4: Evanescent field decrease with increasing frequency (a), and the corresponding effect on S -parameters (b) with a silicon rod waveguide loaded with a thin film of $R_s = 200 \Omega/\text{sq}$.

Comment 2

The reflection results from the 14nm sample are very odd, and the authors claim this could be due to suboptimal alignment or experimental error. I suggest discarding these results, since the errors are almost certainly due to errors rather than intriguing physics. As it is now, those results hinder the readability of the figures.

Response: Thank you for the comment.

We have repeated the measurements for samples displaying anomalies, and we have updated Fig. 2 - Fig. 3 of the original manuscript (as well as all data that follows from them), thereby improving both readability and accuracy. Please also see our discussion in Comment 3 of Reviewer 1 for an explanation of anomalous peaks appearing in certain frequencies.

Comment 3

The paper can be hard to follow at times. The samples are labelled by their optical transmission, which are mapped to approximate thicknesses. They are then referred to in text by the approximate thicknesses, but labelled in figures by their optical transmissions. So on this line “The 14 nm thick film has higher reflections than the other samples for both 6 mm and 12 mm lengths” the reader needs to look at what optical transmission those correspond to first, then link that to the legend in the figures.

Response: Thank you for the comment.

You are right that this is confusing to the reader and should be avoided. Therefore, in the revised document we refer to all samples by their optical transmittance % everywhere (e.g. SWCNT98%), with the corresponding thickness only used as auxiliary in parentheses where deemed necessary.

Comment 4

The Methods sections describes a range of metrology conducted (SEM, TEM, Raman spectroscopy), without any shown results. Taking for example Raman spectroscopy, how was this used to assess film quality? This would be useful in supplementary information.

Response: Thank you for the comment.

We would like to point out that Fig. 1 in the main manuscript presents the representative structural characterization data (SEM, TEM, UV-Vis-NIR, Raman spectroscopy) for a typical SWCNT film, and a discussion follows in Section 2.1, confirming material identity, the low concentration of defects and an isotropic percolating network. All films were prepared in a single batch using established protocols, while comprehensive structural characterization of the films has been carried out in previous publications [2], [3] that are beyond the scope of the present work, which targets THz dielectric waveguide terminations on a device-level.

- [2] E. M. Khabushev, D. V. Krasnikov, J. V. Kolodiaznaia, A. V. Bubis, and A. G. Nasibulin, "Structure-dependent performance of single-walled carbon nanotube films in transparent and conductive applications," *Carbon*, vol. 161, pp. 712–717, 2020.
- [3] E. M. Khabushev, D. V. Krasnikov, O. T. Zaremba, A. P. Tsapenko, A. E. Goldt, and A. G. Nasibulin, "Machine learning for tailoring optoelectronic properties of single-walled carbon nanotube films," *The journal of physical chemistry letters*, vol. 10, no. 21, pp. 6962–6966, 2019.

Concluding Response. Thank you for your valuable comments and for helping us strengthen our manuscript.